# *Plasmodium falciparum* formins are essential for invasion and sexual stage development

Sophie Collier[1,2], Emma Pietsch [1,2], Madeline Dans[3], Dawson Ling [3], Tatyana A. Tavella[1,2], Sash Lopaticki[4,5], Danushka S. Marapana[5], Mohini A. Shibu[1,2], Dean Andrew[1,2], Snigdha Tiash[1,2], Paul J. McMillan [6], Paul Gilson [3], Leann Tilley [1,2] & Matthew W. A. Dixon [4,5 ✉]

The malaria parasite uses actin-based mechanisms throughout its lifecycle to control a range of biological processes including intracellular trafficking, gene regulation, parasite motility and invasion. In this work we assign functions to the *Plasmodium falciparum* formins 1 and 2 (FRM1 and FRM2) proteins in asexual and sexual blood stage development. We show that FRM1 is essential for merozoite invasion and FRM2 is required for efficient cell division. We also observed divergent functions for FRM1 and FRM2 in gametocyte development. Conditional deletion of FRM1 leads to a delay in gametocyte stage progression. We show that FRM2 controls the actin and microtubule cytoskeletons in developing gametocytes, with premature removal of the protein resulting in a loss of transmissible stage V gametocytes. Lastly, we show that targeting formin proteins with the small molecule inhibitor of formin homology domain 2 (SMIFH2) leads to a multistage block in asexual and sexual stage parasite development.

[1] Department of Biochemistry and Molecular Biology, University of Melbourne, Parkville, VIC 3010, Australia. [2] Bio21 Molecular Science and Biotechnology Institute, University of Melbourne, Parkville, VIC 3010, Australia. [3] The Macfarlane Burnet Institute for Medical Research, 85 Commercial Road, Melbourne, VIC 3004, Australia. [4] Department of Infectious Diseases, Doherty Institute, University of Melbourne, Parkville, VIC 3010, Australia. [5] Walter and Eliza Hall Institute, 1G Royal Parade, Parkville, VIC 3052, Australia. [6] Biological Optical Microscopy Platform, University of Melbourne, Parkville, VIC 3010, Australia. ✉email: matthew.dixon@unimelb.edu.au

Malaria remains one of the most widespread infectious diseases, causing an estimated 250 million cases and more than 620,000 deaths in 2021[1]. *Plasmodium falciparum*, a causative agent of malaria, has a complex lifecycle with stages in both the human host and the mosquito vector. Transition between different stages involves the formation of three different motile invasive forms, called the merozoite, the ookinete and the sporozoite, and an elongated but non-motile gametocyte. Invasion and motility are powered by an unusual actin–myosin motor complex that is supported by a double-membrane structure called the inner membrane complex (IMC). Collectively, this structure is called the glideosome[2].

Actin is a ubiquitous protein in eukaryotes and is essential for a range of biological processes, including endocytosis, cell division, maintenance of morphology, intracellular transport, gene regulation and cell motility[3,4]. *P. falciparum* expresses two actin isoforms, actin-I and actin-II, which share less than 80% sequence similarity both with each other and with mammalian actins[4–6]. Unlike mammalian actins, which form long, stable filaments under physiological conditions, actins from Apicomplexan parasites such as *Toxoplasma* and *Plasmodium* remain largely depolymerised. In the absence of actin-stabilising proteins, they form short, unstable filaments that undergo rapid treadmilling[4,6,7].

Actin-I is expressed in all *P. falciparum* lifecycle stages, while actin-II is expressed only in mosquito stages and in gametes. Actin-I is essential for parasite viability and is involved in a range of processes during blood-stage development, including endocytosis, haemoglobin uptake, vesicular trafficking, ring-stage morphology and spatial regulation of chromatin within the nucleus[8–11]. Studies utilising a conditional knockout approach showed that actin-I is needed for *P. falciparum* apicoplast segregation, cytokinesis and formation of daughter merozoites, as well as deformation of the host RBC and merozoite motility during invasion[12]. Similarly, studies utilising the actin modulators jasplakinolide and cytochalasin D show that Actin-I is crucial for gliding motility in ookinetes and sporozoites[13–15]. Actin-I has also been shown to be present in *P. falciparum* gametocytes, where it forms an actin cytoskeleton at the apical tips of the elongated parasite. Disruption of actin, using cytochalasin D, leads to reorganisation of the mitochondrion[16].

Actin-II is unique to *Plasmodium* species and is expressed exclusively in mosquito and sexual stages[5,14–17]. In *P. berghei*, actin-II has been shown to localise to the nucleus in male gametocytes and zygotes, where it plays a critical role in male exflagellation (formation of microgametes) and meiosis. Disruption of the *actin-II* gene in *P. berghei* leads to the inhibition of gametocyte egress and axoneme function, two critical steps in the maturation of male gametes[15]. Complementation of Actin-II knockout parasites with a chimeric version of Actin-I rescues these defects in exflagellation[4]. However, ookinete conversion and oocyst formation is still severely reduced in these mutants. While some oocysts are formed, these remain small and cease to develop past day 8 of infection. This is accompanied by a loss of sporogony and therefore a complete block in parasite transmission[17,18]. Interestingly, methylation of histidine 73 in Actin-II is also required for normal oocyst development and sporogony, but not for male gametogenesis[18].

Apicomplexan parasites encode a surprisingly small set of actin-binding proteins and lack actin-nucleating factors such as the Arp2/3 and WAVE/WASP complexes. The only identified actin-nucleating proteins in *Plasmodium* are the formin family of proteins[19]. Formin proteins are actin-nucleating proteins that are known to regulate actin filament assembly and growth in eukaryotes[20–23]. Formins are important to a wide range of biological processes, including cell adhesion and migration, cell polarity, cytokinesis, endocytosis, cytoskeletal assembly and organisation and cell morphogenesis[23]. Formins are large, multidomain proteins characterised by the presence of the formin homology 1 (FH1) and formin homology 2 (FH2) domains[20]. The FH2 domain forms a homodimer that binds to the barbed end of nucleated actin filaments to promote elongation and to prevent binding of capping protein. The adjacent proline-rich FH1 domain is responsible for binding profilin-actin complexes to facilitate the efficient incorporation of new G-actin monomers onto a growing actin filament[21,22,24].

*P. falciparum* encodes two cytoplasmic formins, known respectively as formin-1 (FRM1) and formin-2 (FRM2), as well as a single nucleus-associated formin-like protein called MISFIT[22,25]. Both formin proteins are large in size (>300 kDa) and contain the characteristic FH1 and FH2 domains. FRM1 contains two additional N-terminal tetratricopeptide repeat domains (TPR), whilst FRM2 has a single N-terminal PTEN-C2-like domain[26]. Both cytosolic formins are expressed during the asexual blood cycle of *P. falciparum* development. FRM1 accumulates at the apical tip of free merozoites, co-occurring with the apical polar rings of microtubules. Interestingly, during invasion, FRM1 comigrates with the moving junction at the merozoite-RBC interface, suggesting that FRM1 may play key roles in actin filament formation and actomyosin motor function[22]. Several attempts to disrupt the FRM1 gene have been unsuccessful, indicating that it is essential for parasite viability[22]. A recent study suggested that FRM2 is located adjacent to apicoplasts in blood-stage *P. falciparum* schizonts and is required for both efficient apicoplast inheritance and merozoite formation[26].

*Toxoplasma gondii* encodes two well-conserved formin proteins, *Tg*FRM1 and *Tg*FRM2, alongside a third divergent formin protein exclusive to coccidians, *Tg*FRM3[22,27,28]. *Tg*FRM1 and *Tg*FRM2 are involved in host cell invasion and gliding motility, with distinct roles in egress and apicoplast segregation, respectively[26,27,29]. *Tg*FRM3 localises to the basal pole of the parasite residual body, where it contributes to intravacuolar cell–cell communication[29].

In this work, we show that formins play essential roles at different stages of *P. falciparum* development. We show that FRM1 is essential for parasite invasion and that FRM2 is essential for gametocyte elongation. We show that FRM2 plays a critical role in modulating and stabilising the actin and microtubule cytoskeletons during development. Lastly, we show that targeting formin proteins with the experimental compound SMIFH2 results in both asexual and sexual stage development arrest. This highlights the possibilities of developing multistage disease and transmission blocking compounds working on actin-related processes within the parasite.

## Results

**The *P. falciparum* formins have distinct cellular locations during asexual blood-stage development.** To investigate the locations of *P. falciparum* FRM1 (PF3D7_0530900) and FRM2 (PF3D7_1219000), we generated transgenic cell lines expressing HA-tagged FRM1 and GFP-tagged FRM2 (Supplementary Fig. 1). For functional studies, we generated a diCRE-mediated conditional knockout[30] of FRM1 in the NF54 background, incorporating flanking loxP sites and a 3xHA tag at the C-terminus (Supplementary Fig. 1a–d). For FRM2, we generated a 3D7-FRM2 knock-sideways line using the selection-linked integration approach[31], incorporating FKBP and GFP tags into the 3' end of the endogenous locus (Supplementary Fig. 1e–g). Correct integration of the FRM1 and FRM2 plasmids into the wild-type loci was confirmed by PCR (Supplementary Figs. 1c, d, g and 9).

Immunofluorescence microscopy of asexual blood-stage parasites revealed a distinct punctate staining pattern in schizont-

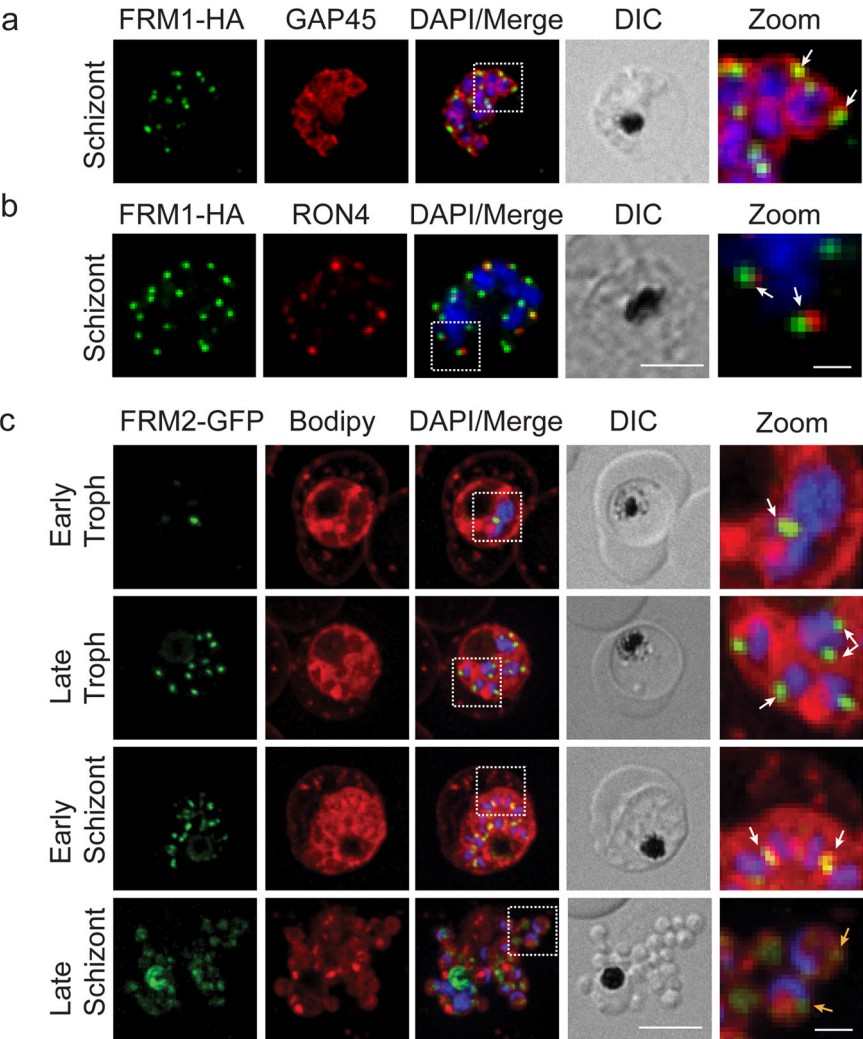

**Fig. 1 FRM1 and 2 show different cellular locations in asexual stage parasites. a** Immunofluorescence microscopy of FRM1-HA parasites labelled with anti-HA (green) and anti-GAP45 (red). **b** anti-HA (green) and RON4 (red). DAPI (blue), merge and Differential interference contrast (DIC) images are shown. Scale bar = 5 μm. Dotted white line outlines the zoom panel. Zoom scale bar = 1 μm. **c** Live-cell immunofluorescence of FRM2-GFP (green) co-labelled with BODIPY-TR-Ceramide (Bodipy, red). DAPI (blue), merge and DIC images are shown. The dotted white line outlines the zoom panel. Zoom Scale bar = 1 μm.

stage parasites with individual FRM1-HA labelled puncta associated with the developing merozoites, as outlined by labelling with the inner membrane complex (IMC) marker GAP45 (Fig. 1a, zoom, white arrows). Previous work has shown that FRM1 locates at the apical end of the merozoite[22]. Using antibodies to the apically located protein RON4, we confirmed that FRM1-HA locates close to the RON4 puncta at the apical end of the merozoites (Fig. 1b, zoom, white arrows).

Live-cell microscopy of FRM2-GFP counterstained with the lipid dye BODIPY-TR ceramide and the nuclear stain DAPI, revealed a different pattern of localisation for FRM2. FRM2-GFP fluorescent puncta locate adjacent to the nucleus across asexual development (Fig. 1c, zoom, white arrows). Throughout development, some additional labelling can be seen at the digestive vacuole (Fig. 1c). Staining with BODIPY and DAPI reveals nuclei of the dividing parasites, each associated with puncta (Fig. 1c). In free merozoites, a single punctum is associated with each merozoite (Fig. 1c, yellow arrows). This labelling pattern is equivalent to that previously shown for HA-tagged FRM2[26]. The expression of the GFP-tagged FRM2 was confirmed by Western blot (Supplementary Figs. 1h and 9). Given the perinuclear location of FRM2-GFP and reports that some mammalian

formins locate at microtubule organising centres (MTOC), we performed IFAs with antibodies recognising centrin-3 which locates at the MTOC. These experiments show a high degree of overlap between the FRM2 staining and centrin, with some additional regions of FRM2-GFP labelling within the parasite (Supplementary Fig. 1i).

**FRM1 is required for parasite invasion**. We made use of the FRM1-HA diCRE conditional knockout (cKO) line to investigate gene function. A loxP site was positioned within an artificial intron that was placed within the gene sequence corresponding to the FH2 homology domain of FRM1, and a second loxP site was placed downstream of the HA tag. Following addition of Rapalog, excision is expected to truncate the C-terminal FH2 domain of FRM1 (Supplementary Fig. 1d). Tightly synchronised ring-stage parasites (0–3 h post invasion (hpi)) were treated with or without Rapalog. Schizont-stage parasites were harvested, DNA extracted and PCR analysis was performed confirming that FRM1 had been deleted (Supplementary Figs. 1d and 9). Likewise, IFAs were performed on schizont-stage parasites confirming a loss of the characteristic FRM1-HA puncta in the Rapalog-treated group (Fig. 2a). We were unable to detect the 312 kDa FRM1-HA by

western blotting, due to low expression levels. Deletion of FRM1 had no effect on the labelling pattern of GAP45, RON4, EBA175 or AMA1 (Fig. 2a and Supplementary Fig. 2a–c), confirming that merozoites had developed within the schizonts.

To investigate the location of actin following FRM1 deletion, we transfected the FRM1-HA parasite line with an actin chromobody reporter fused to GFP. We performed live-cell microscopy on DMSO and Rapamycin-treated cells. In controls, the actin chromobody is observed as bright puncta at the apical end of the merozoites with some additional background fluorescence, consistent with previously reported data (Supplementary Fig. 2d)[26]. Deletion of FRM1 leads to loss of apical staining and redistribution of the reporter to the cytoplasm, with labelling of bright filamentous structures that interweave between the merozoites (Supplementary Fig. 2d). These data confirm that deletion of FRM1 leads to a change in actin localisation.

Parasites were next assessed for their ability to reinvade RBCs. Conditional KO of FRM1 led to a 95% reduction in parasitemia in the following cycle (Fig. 2b). The same Rapalog treatment of the parent (NF54) parasites had a small but non-significant effect on invasion (Supplementary Fig. 2e). These data suggest that FRM1 is either required for merozoite egress from the schizont or invasion into the RBC. To investigate this, we used live-cell imaging of the merozoite egress and invasion process.

Synchronous FRM1-HA late schizonts that had received DMSO at ring-stage were mounted in an environmental chamber for live-cell imaging. Fields containing schizonts in which internal merozoites could be discerned, and which appeared rounded up and had highly condensed hemozoin were selected and imaged using low-intensity brightfield illumination at four frames per second[32,33]. Eleven schizont egress events were observed for DMSO-treated FRM1-HA parasites, allowing visualisation of 56 merozoite contacts of ≥1 s with neighbouring RBCs, resulting in 16 RBC invasions (Fig. 2c). After initial RBC contact, the merozoite usually deforms its target RBC for several seconds before commencing to penetrate the RBC (Supplementary Video 1). This pre-invasion phase of FRM1-HA parasites had a mean duration of 26.1 s, which is longer than the mean of 10 s observed in other parasite strains used in previous studies (Fig. 2c)[33]. Merozoite internalisation in which the parasite completely embeds itself in the RBC usually takes about 10 s, similar to the mean of 8.4 s observed here. About 30 s after internalisation is complete, the target RBC begins to curl at the edges and develop a spikey appearance called echinocytosis, similar to that observed here (Fig. 2c)[33].

Having confirmed that most of the invasion kinetics parameters for the FRM1-HA parasite were similar to those observed for other parasite lines, ring-stage FRM1-HA parasites were treated with rapamycin to activate diCRE-mediated deletion. Schizont-stage parasites were prepared for imaging as described above. From seven egressing rapamycin-treated schizonts, we observed no merozoite invasion, which was significantly less than the number observed for DMSO-treated parasites (mean of 1.6 invasions per egress; Fig. 2d). As invasion success is dependent on the number of RBCs that are near the schizont, the number of merozoite-RBC contacts per egress was measured (Fig. 2e). The mean number of contacts is significantly higher for DMSO-treated (5.1 per egress) than rapamycin-treated (2.1 per egress) schizonts. Close examination of the invasion videos revealed that merozoites in the rapamycin-treated schizonts more frequently remain anchored to the hemozoin-containing residual body for tens of seconds following egress (Supplementary Videos 2 and 3). The attachment of merozoites to the residual body was not the sole cause of reduced invasion for the rapamycin-treated free merozoites, as those merozoites that contacted RBCs failed to invade, while the DMSO-treated merozoites invaded with a mean efficiency of 39% (Fig. 2f).

Close examination of the rapamycin-treated FRM1-HA merozoites indicated they only deformed their target RBCs very weakly if at all. In this respect, they resemble merozoites treated with cytochalasin D (Supplementary Videos 2 and 3)[33]. Unlike cytochalasin-D treatment, however, the rapamycin-treated FRM1-HA target RBCs rarely underwent echinocytosis, suggesting FRM1 may be required for the formation or release of apical secretory organelles such as micronemes or rhoptries[33].

### Knock sideways of FRM2 leads to a growth defect.

Previous work reported a diCRE conditional knockout of FRM2 that showed a marked growth defect due to an effect on cytokinesis[26]. To enable finer dissection of the cellular defects, here we made use of the fact that the FRM2-GFP parasite line contains two FK506-binding protein (FKBP) domains on each side of the GFP (Supplementary Fig. 1f). This line was co-transfected with a mislocaliser plasmid expressing an FKBP-rapamycin binding (FRB) domain, fused to an mCherry reporter and a plasma membrane targeting signal (LYN) (Supplementary Fig. 1j). The LYN plasma membrane mislocaliser was chosen as it differs from the location of FRM2-GFP which is adjacent to the nucleus. In the presence of Rapalog, the FKBP and FRB domains dimerise, and the FRM2-GFP protein is targeted to the parasite membrane (Supplementary Fig. 1k). Following Rapalog treatment, we see co-occurrence of the FRM2-GFP and mislocaliser-mCherry fluorescence signals at the parasite periphery, confirming knock sideways of FRM2-GFP to the plasma membrane (Fig. 2g). Using knock sideways we see a less dramatic effect on growth than the previously reported FRM2 knockout defect[26], with a 38% reduction in parasite growth after two cycles (Fig. 2h). The discrepancy between the published conditional KO data and our knock-sideways data is most likely due in part to the incomplete misslocalisation of FRM2-GFP in the parasites, with some protein still appearing to make it to the site of function in asexual stage parasites. Treatment of the parent (3D7) parasites with Rapalog for the same duration had no effect on parasite growth (Supplementary Fig. 2f).

To better understand the mechanism underlying the reduced growth rate of parasites following mislocalisation of FRM2, we investigated whether nuclear division was affected. Centrin proteins locate to the microtubule organising centres of dividing parasites and play key roles in mitotic division. We performed IFAs and examined the co-occurrence of FRM2-GFP and centrin following knock sideways (Supplementary Fig. 2g). We observed a significant reduction in the number of nuclei and the number of FRM2-GFP puncta consistent with a defect in cell division (Supplementary Fig. 2h). Examination of the IFAs show that mislocalisation of FRM2-GFP leads to a reduction in the level of FRM2-GFP and centrin colocation. We observed a significant decrease in the Pearsons co-efficient from $0.72 \pm 0.09$ in the control to $0.61 \pm 0.1$ in the Rapalog-treated group (1 equals complete colocation, Supplementary Fig. 2i). IFAs using antibodies to apicoplast marker ACP were also performed to assess apicoplast morphology and inheritance following mislocalisation of FRM2-GFP (Supplementary Fig 2j). Apicoplast segregation appeared to be largely unaffected; however, in some cells an abnormal, clumped morphology of the apicoplast was observed.

### FRM1 and FRM2 exhibit different cellular locations in gametocytes.

Immunofluorescence microscopy was used to determine the location of FRM1 in gametocytes. FRM1-HA exhibits a diffuse punctate labelling pattern in the parasite cytoplasm in all stages of gametocyte development. In contrast, β-tubulin labelling is restricted to the parasite periphery in developing gametocytes

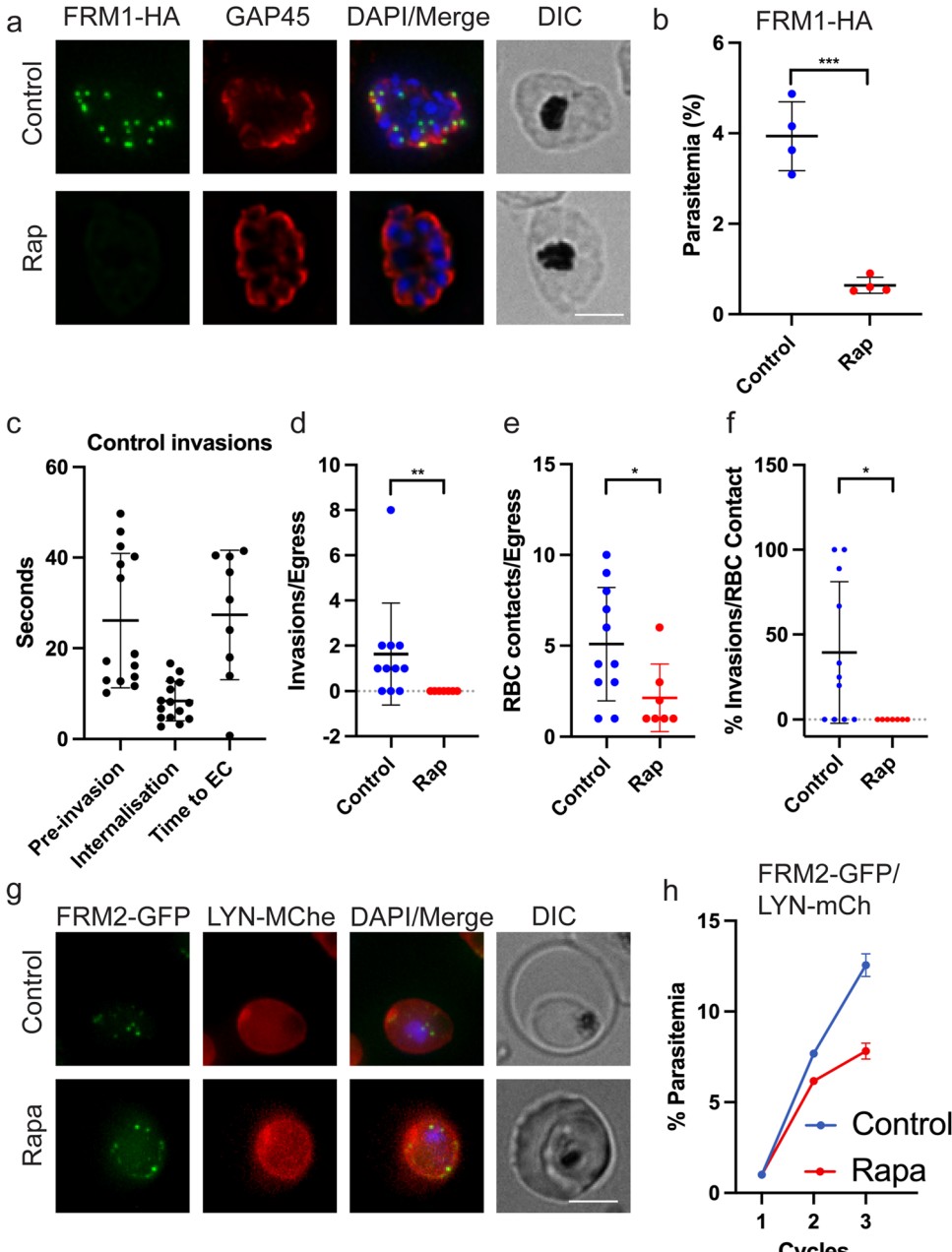

**Fig. 2 FRM1 is needed for invasion while FRM2 is required for division. a** Immunofluorescence microscopy of control and rapamycin (Rap) treated FRM1-HA parasites labelled with anti-HA (green) and anti-GAP45 (red). Merge, DAPI (blue) and DIC images are also shown. Scale bars = 5 μm. **b** Asexual growth assay of FRM1-HA parasites following treatment with rapamycin (Rap) or DMSO (Control). The mean values and standard deviation from four experiments are plotted. An unpaired $t$ test was performed, ***$P < 0.001$. **c**–**f** Rapamycin-mediated deletion of FRM1 blocks merozoite invasion of human red blood cells (RBCs). **c** Merozoites from FRM1-HA schizonts mock-treated with DMSO appear to invade RBCs with similar kinetics, with respect to internalisation and time to echinocytosis (EC), as other parasite lines; but were found to have a longer pre-invasion phase. **d** Following treatment with 100 nM Rap from ring stage, the merozoites liberated from egressing schizonts do not reinvade. **e** Merozoites released from Rap-treated schizonts make similar numbers of contacts (>1 s) with neighbouring RBCs as mock-treated merozoites. **f** Rap-treated merozoites invade significantly less efficiently than DMSO-treated merozoites. Individual data points from each cell analysed is plotted. Eight and 11 egress events were scored for the Control and Rapamycin-treated groups, respectively. The mean and standard deviation are shown. A Mann–Whitney test was performed, *$P < 0.05$, **$P < 0.01$. **g** Live-cell images of asexual stage transfectants. FRM2-GFP (green) puncta can be seen adjacent to the nuclei in control-treated cells. Following treatment with Rapalog (Rapa), puncta are located at the parasite membrane. Scale bar = 5 μm. **h** Long-term growth assay. Parasites were treated from ring stage in cycle 1 and parasitemias measured at the ring stage after 1 and 2 cycles. The data are plotted as the accumulative parasitemia over the three cycles of growth, following Rapa or Control treatment. The mean and standard error of the mean are shown, from three experiments. Unpaired $t$ tests were performed at each cycle. $P < 0.0001$ for both cycle 2 and 3 Control and Rapa comparisons. Source data for this figure is provided in the Source Data files.

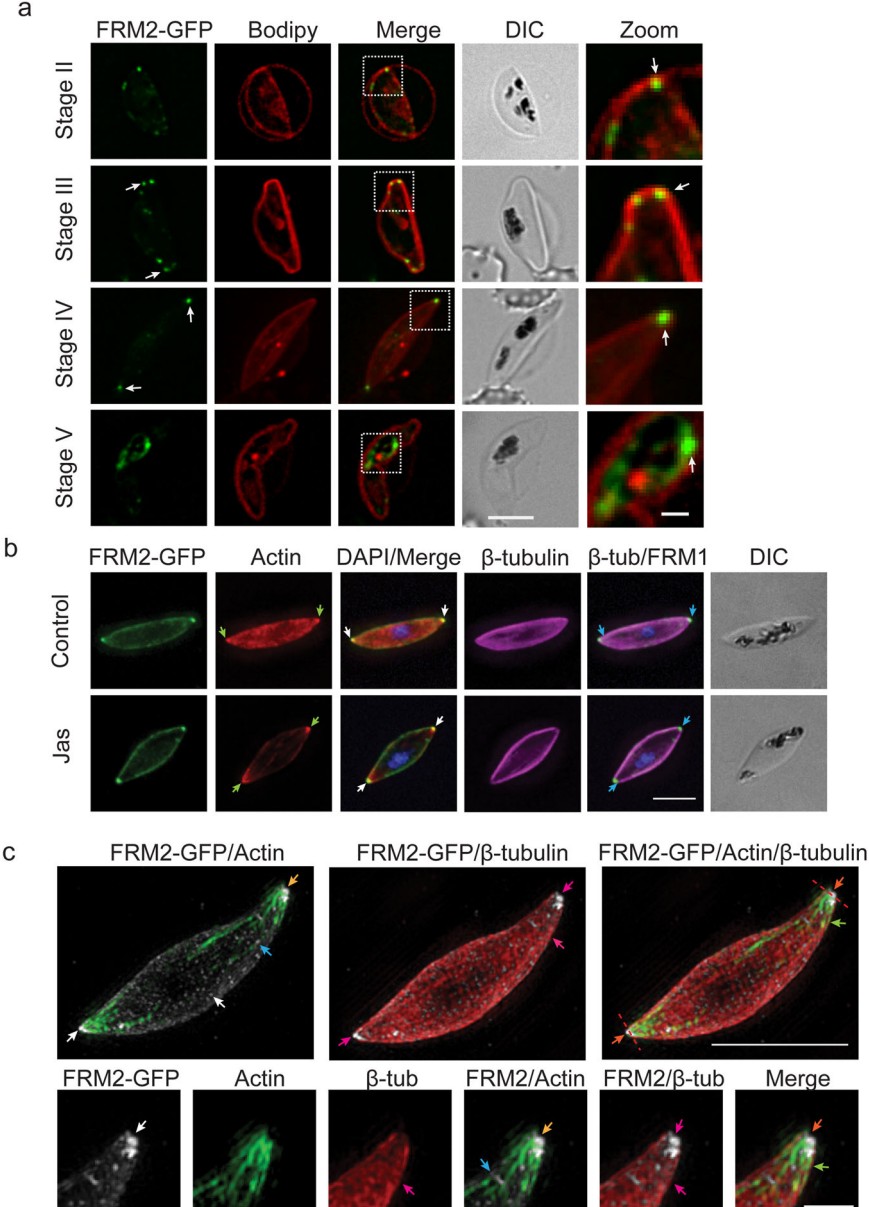

**Fig. 3 FRM2-GFP locates with actin and tubulin at the ends of the developing gametocytes. a** Live-cell microscopy of stage II–V FRM2-GFP (green) gametocytes co-stained with BODIPY-TR-Ceramide (Bodipy, red). Arrows indicate the distinct puncta at the periphery of early-stage gametocytes and at the tips of the stage IV gametocyte. Scale bars = 5 μm. Dotted white line outlines the zoom panel. Zoom scale bar = 1 μm. **b** Immunofluorescence microscopy of stage IV gametocytes triple-labelled with anti-GFP (green), anti-actin (red), anti-β-tubulin (magenta) and DAPI (blue). Gametocytes were incubated with DMSO (Control) and Jasplakinolide (Jas). Scale bars = 5 μm. **c** 3D-structured illumination microscopy of stage IV gametocytes labelled with anti-GFP (grey), anti-actin (green) and anti-β-tubulin (red). Imaging was performed on Jas-treated parasites. Merged images of FRM2-GFP/actin, FRM2-GFP/β-tubulin and all three channels are shown. Zoomed images of the gametocyte tip showing each individual channel and different combinations of the three together. Scale bar = 5 μm. Zoom scale bar = 1 μm. Red dotted lines in (**b**) indicate the lines used for the fluorescence intensity plots of each channel shown in Supplementary Fig. 4a. Additional examples can be found in Supplementary Fig. 4.

before the disassembly of the microtubules in stage V of development (Supplementary Fig. 3a).

In contrast to FRM1-HA, live-cell microscopy of FRM2-GFP shows distinct puncta at the periphery (as marked by BODIPY-TR staining) of stage II and III gametocytes (Fig. 3a, white arrows), with a weaker population in the cytoplasm (Fig. 3a). In stage IV gametocytes, the puncta relocate to distinct foci at the pointed ends of the gametocyte close to the membrane (Fig. 3a, white arrows). This membrane association of FRM2-GFP was confirmed by immunofluorescence assays using the IMC marker PhIL1 (Supplementary Fig. 3b). Following the transition of the

parasite to stage V of development, this tip labelling profile is lost, with the majority of the fluorescence signal relocating to a region close to the digestive vacuole (Fig. 3a).

Our previous work revealed an accumulation of actin at the tips of stage IV gametocytes, where it overlaps with the microtubule cytoskeleton[16]. To investigate if FRM2-GFP co-occurs with the actin and tubulin cytoskeleton, we performed immunofluorescence assays using antibodies to actin and β-tubulin (Supplementary Fig. 3c, d). Antibodies to actin show an accumulation of protein at the tips of the stage IV gametocyte, which co-occurs with the FRM2-GFP puncta observed at the tips,

consistent with the role of formins in actin filament formation (Supplementary Fig. 3c). We next investigated the location of FRM2-GFP relative to the microtubule network. As previously described, labelling with anti-β-tubulin shows a characteristic peripheral staining pattern with the microtubules spanning almost to the end of the stage IV gametocytes (Supplementary Fig. 3d, arrow). Dual labelling with anti-GFP and β-tubulin antibodies shows that FRM2-GFP sits right at the tips of the microtubule network (Supplementary Fig. 3d).

**The arrangement of FRM2, microtubule and actin networks at the tips of stage IV gametocytes.** The dual labelling experiments with actin, tubulin and FRM2-GFP suggest an association of the cytoskeletons with FRM2. To investigate this in more detail we performed triple antibody labelling of gametocytes, with and without actin stabilisation with Jasplakinolide (Jas). To achieve the triple labelling, we used chicken anti-GFP, rabbit anti-actin and mouse anti-β-tubulin. As observed by live-cell microscopy, FRM2-GFP accumulates at the gametocyte tips, with and without Jas-stabilisation (Fig. 3b, white arrows). Actin is also concentrated at the tips, a profile that is enhanced by treatment with Jas (Fig. 3b, green arrows). Anti-β-tubulin antibodies label the microtubule network that lies at the periphery of these stage IV gametocytes. Interestingly, β-tubulin labelling is weaker at the tips where FRM2-GFP was localised, and instead appears to sit around the periphery of FRM2-GFP puncta (Fig. 3b, blue arrows). As previously described[16], we see regions of overlap between the actin and β-tubulin labelling.

To analyse the protein organisation at the gametocyte tips at a higher resolution, we used 3D-Structured Illumination Microscopy (3D-SIM). 3D-SIM can achieve a resolution of ~ 125 nm (x and y), giving an eightfold increase in volume resolution, compared with conventional widefield microscopy. 3D-SIM imaging of triple-labelled gametocytes revealed the FRM2-GFP signal as two elongated puncta at the apical tips, as well as in many smaller puncta distributed throughout the cytoplasm (Fig. 3c, white arrows). The larger apical FRM2-GFP puncta appear to be interwoven with actin filaments/bundles (Fig. 3c, yellow arrows). The smaller FRM2-GFP puncta are observed in close proximity to the actin bundles that radiate from the gametocyte tips (Fig. 3c, blue arrows). The actin bundles at the apical tips also overlap with the microtubule network (Fig. 3c, green arrows), which wraps around the cell, but is largely excluded from the region of the FRM2-GFP puncta at the gametocyte tips (Fig. 3c, pink arrows). Close examination of zoomed images reveals β-tubulin staining around both FRM2-GFP and actin at the tips, with some overlap observed between the three components (Fig. 3c, orange arrow). To investigate this arrangement in more detail, we used a line scan analysis to profile the distribution of each component at the gametocyte tips (Supplementary Fig. 4a). The β-tubulin appears to surround (and partly overlap with) a core that contains actin and FRM2-GFP (Supplementary Fig. 4a). Additional examples can be found in Supplementary Fig. 4b, c.

**Formins are required for correct gametocyte development.** We next investigated if the formin proteins play a role in gametocyte development. FRM1-HA parasites were committed to gametocyte development and treated with and without Rapalog. The morphology and development of the gametocytes were monitored by Giemsa smears, and the stage distribution and parasitemia were recorded at days 4, 6, 8 and 10 of development. There is a small decrease in the overall parasitemia by day 8 of development in the Rapalog-treated (0.56 ± 0.29%) compared with the control group (0.93 ± 0.25%). Analysis of the stage distribution across

development shows an apparent delay in development from day 6 onwards, with an increase in stage III gametocytes (Supplementary Fig. 5a). There is also a small increase in the number of irregular or deformed gametocytes and the large majority of the treated parasites with a stage IV like morphology, appear more swollen than the untreated controls (Supplementary Fig. 5b). Measurements of day 6 gametocytes show a decreased length to width ratio in the treated group (1:2.4) compared to controls (1:3) confirming that the cells are shorter and wider following FRM1 deletion, most probably due to the delay in development (Supplementary Fig. 5c). This delay persists throughout development with only 30% of gametocytes reaching stage V of development by day 10 compared to 63% in the untreated controls (Supplementary Fig. 5a). Treatment of the parent NF54 parasites had no effect on gametocyte development (Supplementary Fig. 5d).

To assess the function of FRM2 in gametocyte development, we co-transfected the FRM2-GFP parasite line with a nuclear mislocaliser (NLS) (Supplementary Fig. 1j). The NLS mislocaliser was chosen as the FRM2-GFP locates preferentially at the parasite periphery throughout gametocyte development. To determine the effects that FRM2-GFP mislocalisation had on gametocyte shape, cells were cultured in the presence or absence of Rapalog continuously from Day 0 through to Day 10, representing stage I to V of gametocyte development. The morphology of the gametocytes was monitored daily by live-cell microscopy and Giemsa-stained smears (Fig. 4a, b). Analysis of live-cell images collected across different stages of development indicated that FRM2-GFP had been successfully mislocalised to the nucleus following Rapalog treatment, as indicated by the overlapping GFP and mCherry fluorescence signals (Fig. 4a and Supplementary Fig. 6a–c). In some parasites, colocation of FRM2-GFP and the mislocaliser can be seen at the cell periphery (Fig. 4a and Supplementary Fig. 6a–c). This may represent protein that had already been delivered to the parasite periphery and was not able to be mislocalised due to its tight association with the membrane. Counts of Giemsa-stained slides of the treated parasites revealed a distinct change in gametocyte morphology from day 4 onwards with more than 50% of the Rapalog-treated gametocytes having a deformed or rounded morphology, compared to 5% in the untreated controls; and the remaining parasites being delayed in development (Fig. 4c and Supplementary Fig. 6a–c). Long-term Rapalog treatment of the FRM2-GFP parasite line without the mislocaliser plasmid had no effect on parasite development or morphology (Supplementary Fig. 6d).

Live-cell imaging revealed that parasite morphologies ranged from crescent to tear drop to a more rounded morphology. The crescent and tear drop-shaped gametocytes retained some FRM2-GFP at the parasite tips, (middle panel, Fig. 4a and Supplementary Fig 6a–c). In contrast, parasites that took on a completely rounded appearance, showed more complete mislocalisation of FRM2-GFP (Fig. 4a and Supplementary Fig. 6a–c, bottom panels). Measurements of gametocytes from treated and untreated groups showed a significant decrease in gametocyte length and a significant increase in parasite width (Supplementary Fig. 6e).

**FRM2 mislocalisation causes the disassembly of the actin and microtubule cytoskeletons in gametocytes.** We examined the effects that FRM2-GFP mislocalisation had on the cellular distribution of the actin and microtubule networks using both widefield (Supplementary Fig. 7) and 3D-SIM immuno-fluorescence microscopy (Fig. 4d). FRM2-GFP and actin exhibit the characteristic tip labelling in untreated gametocytes (Fig. 4d and Supplementary Fig. 7a). Upon FRM2-GFP mislocalisation, some gametocytes exhibited long, stabilised actin filaments (Fig. 4d and Supplementary Fig. 7a, middle panel), which

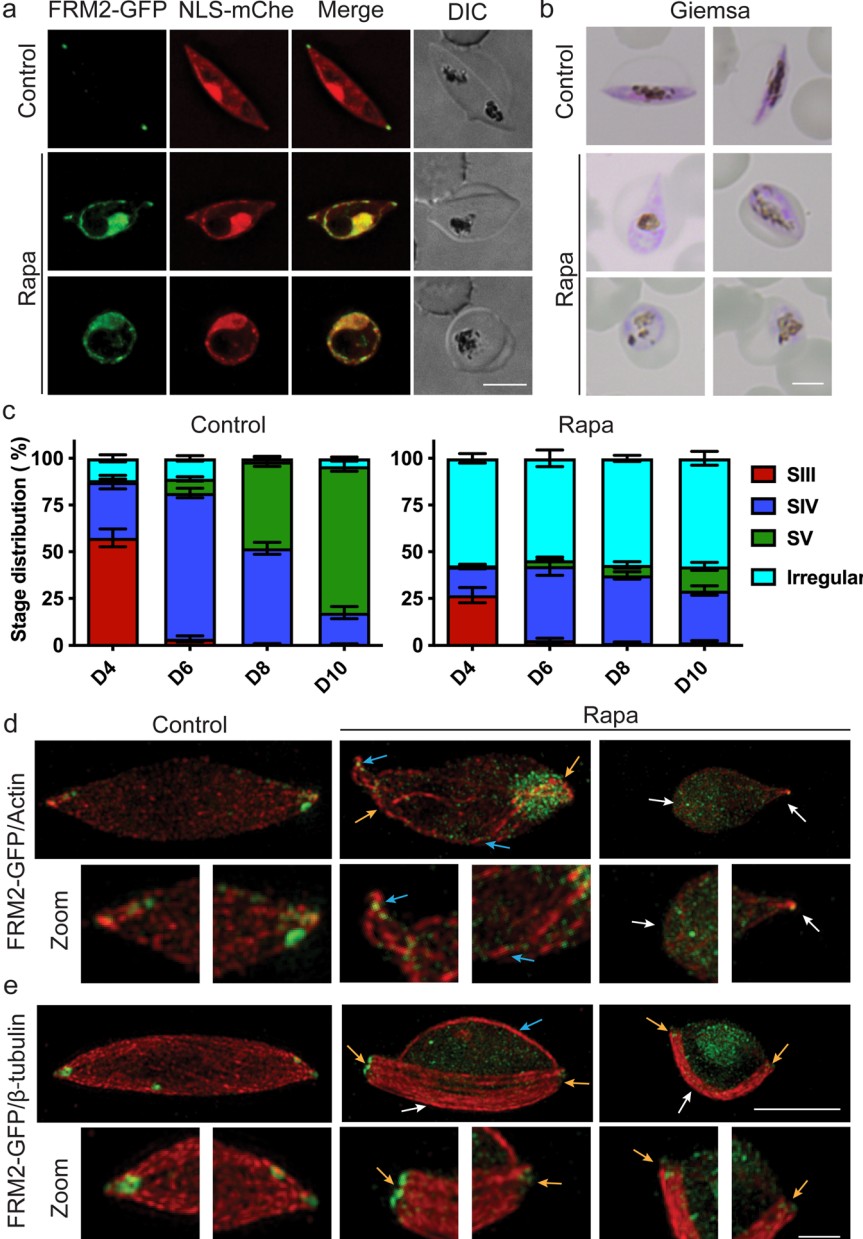

**Fig. 4 Knock sideways of FRM2 leads to an arrest in gametocyte development and disrupted actin and microtubule networks. a** Live-cell microscopy of stage IV gametocytes treated with ethanol (Control) or Rapalog (Rapa). Mislocalisation of FRM2-GFP to the nucleus using the NLS mislocaliser resulted in irregular morphologies including abnormal (middle panel) and rounded morphologies (bottom panel). The nucleus is delineated by the native fluorescence from the mCherry reporter incorporated into the NLS mislocaliser (red). FRM2-GFP (green) is evident at the periphery of the nucleus. Scale bars = 5 μm. **b** Images of Giemsa-stained cells following treatment with and without Rapa. Representative examples of gametocytes from day 7 of the assay when control cells exhibited mostly stage IV morphology. Scale bars = 5 μm. **c** Graphs showing the gametocyte stage distribution for day 4, 6, 8 and 10. Giemsa counts were performed from four separate experiments. Gametocytes were separated based on morphology (stage III–V). Irregular parasites include those that are abnormal or rounded. The means and standard error of the mean are shown from three independent experiments. **d, e** 3D-Structured Illumination Microscopy of stage IV control (ethanol) and Rapalog (Rapa) treated gametocytes. **d** Gametocytes were labelled with anti-GFP (green) and anti-actin (red). White arrows point to short, stabilised actin filaments. Yellow arrows highlight long, stabilised actin filaments. Blue arrows depict FRM2-GFP puncta positioned between actin filaments. **e** Gametocytes labelled with anti-GFP (green) and anti-β-tubulin (red). White arrows highlight the preferential bundling of microtubules to one side of the gametocyte. Blue arrows point to a lone bundle of microtubules present in Rapa-treated gametocytes with an abnormal morphology (middle panel). Yellow arrows depict FRM2-GFP puncta positioned where microtubules terminate. Scale bars = 5 μm. Zoomed images are displayed below each panel. Zoom scale bars = 1 μm. Source data for this figure is provided in the Source Data files.

resembled those observed following Jasplakinolide stabilisation. These actin filaments appeared to radiate from the gametocyte tips where FRM2-GFP puncta remained, often wrapped around the parasite nucleus, and were decorated with FRM2-GFP puncta (Supplementary Fig. 7a). In the tear drop-shaped gametocytes, a

small population of actin is concentrated at the remaining pointed tip (Fig. 4d), while in completely rounded parasites actin is broadly distributed across the cell (Supplementary Fig. 7a).

In Rapalog-treated gametocytes, the development of the microtubule network appears to be altered. The microtubules

are largely restricted to one side of the parasite (Fig. 4e, white arrows), more typical of earlier (stage II/III) gametocytes. The remnant FRM2-GFP puncta at the gametocyte tips remain adjacent to the microtubules, but the microtubules have failed to expand around the gametocytes, thereby affecting the elongation process (Fig. 4e and Supplementary Fig. 7b). In control gametocytes, FRM2-GFP appears as a tight doughnut-shaped structure at one end of the enveloping carapace of microtubules. By contrast, the puncta are more distributed in the treated parasites, including a row of puncta along the end of the microtubule plate. Dual labelling with antibodies recognising the IMC marker, PhIL1, confirms that the IMC is maintained at the cell periphery following the mislocalisation of FRM2-GFP (Supplementary Fig. 7c).

**The small molecule inhibitor, SMIFH2, kills asexual stage parasites.** The small molecule inhibitor of Formin Homology 2 domains (SMIFH2) abolishes formin-dependent actin polymerisation by targeting the FH2 domain of the protein. Previous studies have established that SMIFH2 has profound effects on F-actin cytoskeletal structures in animal cells and fission yeast, with downstream effects on microtubules and the integrity of the Golgi apparatus[34,35]. We assessed the viability of asexual wild-type NF54 parasites treated with SMIFH2 using a standard 48 h growth assay, as previously described[36]. SMIFH2 inhibited growth with an $IC_{50}$ of $24 \pm 6\,\mu M$ (Supplementary Fig. 8a). Analysis of Giemsa images reveals that at higher concentrations ($\geq 50\,\mu M$), growth is completely attenuated and only pyknotic forms are observed, whereas at $25\,\mu M$, parasites appear to develop to schizont stage, but the rate of reinvasion is dramatically impaired, with few rings being observed (Supplementary Fig. 8a, b). These results are reminiscent of the block in invasion observed upon conditional knockout of FRM1 (Fig. 2). Likewise, an early arrest in parasite development is consistent with the replication defect observed in the knock-sideways FRM2-GFP experiments (Fig. 2).

**Treatment of gametocytes with SMIFH2 results in abnormal gametocytes and an altered cytoskeleton.** To assess the effectiveness of SMIFH2 in gametocytes we made use of the FRM2-GFP parasite line. We treated stage IV gametocytes with a range of drug concentrations and performed live-cell immunofluorescence microscopy to determine the effects treatment has on parasite shape and FRM2-GFP localisation. FRM2-GFP parasites were treated with a range of SMIFH2 concentrations from $12.5-100\,\mu M$. We observe a dose-dependent response with an increasing number of cells which have mislocalised FRM2-GFP and abnormal morphologies when compared to controls (Fig. 5a, b). Immunofluorescence assays were performed using anti-$\beta$-tubulin, anti-actin antibodies and native FRM2-GFP fluorescence (Fig. 5c). SMIFH2 inhibition resulted in decreased actin tip staining (blue arrows) and altered microtubule labelling (white arrows) in a dose-dependent manner (Fig. 5c). The phenotypes observed in the higher concentrations of SMIFH2 are consistent with the phenotype observed in the FRM2-GFP mislocalisation experiments.

## Discussion

The ubiquitous protein actin plays central roles in controlling a range of biological processes in *Plasmodium*, including invasion, motility, nutrient uptake and gene regulation[8–11]. In this work, we investigated the function of the two cytosolic formin proteins of *Plasmodium falciparum*, showing that they have distinct cellular locations within the asexual and sexual blood stages of development. In addition, we show for the first time that FRM1 is essential for merozoite invasion and that FRM2 is required for

gametocyte maturation, evidently via effects on the actin and microtubule cytoskeletons.

Previous work has shown the importance of actin to merozoite invasion, where it forms the central driving force of the actin–myosin motor complex, which is anchored to the inner membrane complex (IMC) plates[12]. Using a diCRE-mediated conditional knockout approach, we show that deletion of FRM1 blocks invasion at the earliest stage of the process (Fig. 6a). Live-cell examination of deleted parasites showed no defect in schizont rupture, but a decreased efficiency of merozoite escape from the parent cell and ablation of the ability of merozoites to bind to a new RBC and initiate the downstream invasion cascade[33]. This contrasts with data from a conditional knockout of actin-I, where merozoites were reported to bind to new RBCs, secrete microneme proteins and engage the tight junction, but were unable to invade the RBCs[12]. Given these results, it is possible that FRM1 plays additional roles, for example in binding to and anchoring the microtubule networks, which in turn may impact the ability of the parasites to engage the glideosome and invade RBCs[12].

In this study, we show that FRM2 is located adjacent to the nucleus, which is distinct from the location of FRM1. In our studies, mislocalisation of FRM2 to the parasite plasma membrane led to a decrease in parasite replication (Fig. 6b). This is not as pronounced as the diCRE conditional knockout previously reported for FRM2[26]. This suggests that some FRM2 protein is still trafficked to its site of function adjacent to the nucleus in the knock-sideways experiments, resulting in a defect but not a complete loss of parasite viability. This supports our observation that Rapamycin treatment caused only partial mislocalisation in some cells. This may be due to FRM2 already being located at its site of function when the Rapamycin treatment is initiated. Nonetheless, dual labelling experiments with antibodies to centrin, a marker of the microtubule organising centre (MTOC), showed that the FRM2 and centrin puncta locate adjacent to each other. In higher eukaryotes, formin proteins have been shown to be involved in cytokinesis through the organisation and assembly of the contractile ring and the mitotic spindle[37]. Given the reduced parasitemia we observe following mislocalisation of FRM2 it is plausible that FRM2 plays similar roles in parasite cell division. This is consistent with the data presented by Stortz et al. that shows upon conditional knockout of FRM2 the parasites are defective in cytokinesis and are unable to correctly segregate the apicoplast into the daughter cells as they form[26]. This is also observed for *Toxoplasma* formin-2, which is also reported to play roles in apicoplast segregation and cell division[29].

The characteristic shape of the *P. falciparum* gametocyte is driven by the assembly of a dense microtubule network that is supported by the IMC[38–41]. In addition to the microtubule network, the gametocyte also possesses an actin cytoskeleton, with an accumulation of actin at the tips of the stage IV gametocytes[16]. In this study, 3D-SIM imaging reveals a distinct localisation pattern for FRM2 in stage IV gametocytes. We show that FRM2 accumulates at the tips of gametocytes where it interweaves and partially overlaps with both the actin and microtubule networks. Interestingly, several eukaryotic formins have been shown to bind and regulate microtubules via either their N- or C-terminal regulatory domains or directly through the FH2 domain[42]. As *Plasmodium* formins lack the N- and C-terminal domains previously shown to be required for microtubule association, it suggests that *Plasmodium* formins may bind microtubules through their FH2 domains.

The removal of FRM2 from the tips of the gametocytes by mislocalization to the nucleus leads to a dramatic collapse of the actin cytoskeleton and a marked impact on the organisation of the microtubule network. In cells with a rounded appearance, a small patch of microtubules can be seen around one side of the

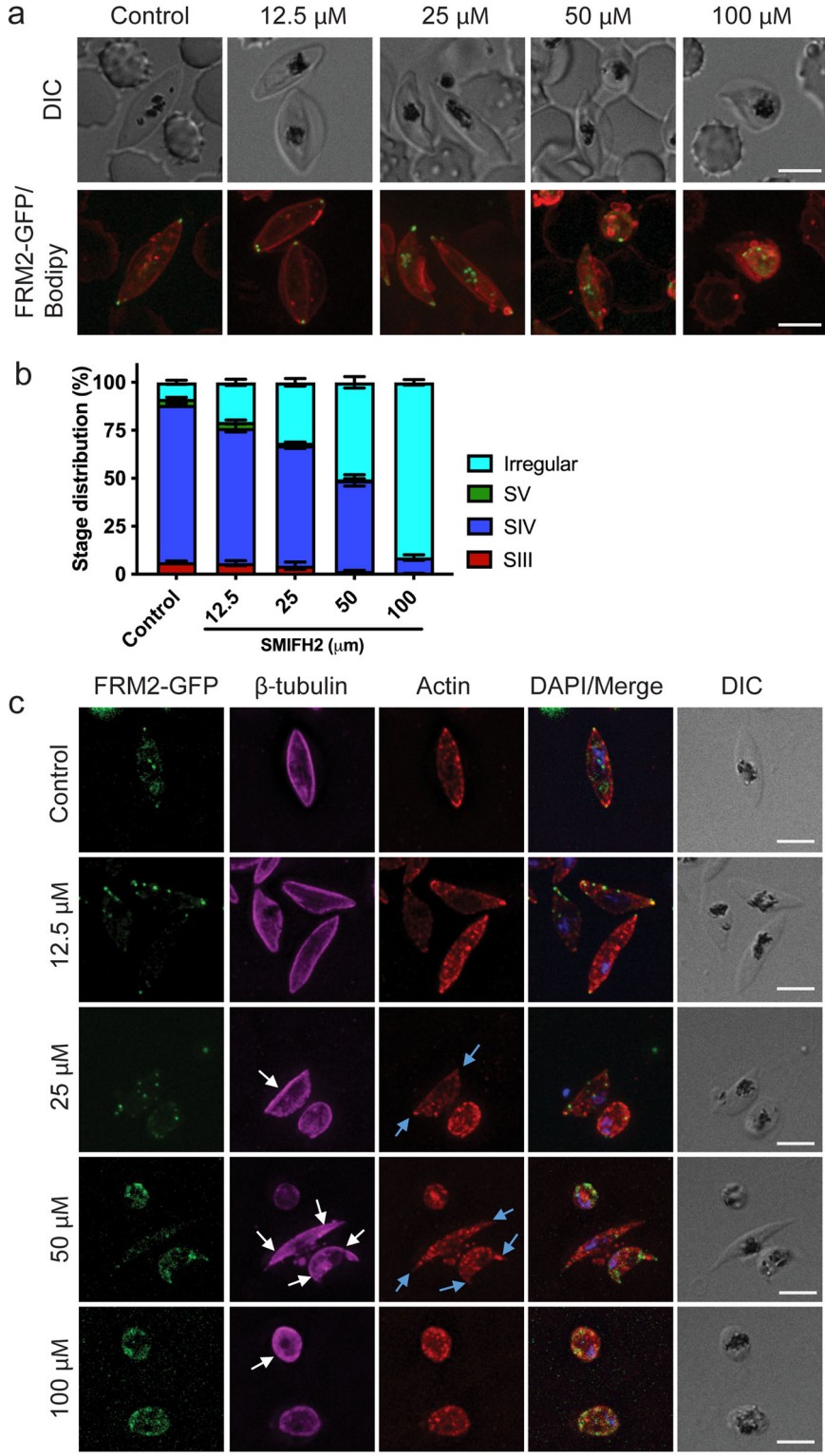

**Fig. 5 The small molecule inhibitor SMIFH2 kills asexual stages and gametocytes. a** Live-cell immunofluorescence microscopy of stage IV gametocytes following overnight treatment with SMIFH2 inhibitor at differing concentrations. Merged images of FRM2-GFP (green) parasites labelled with the membrane dye Bodipy (red) and DIC images are shown. Scale bars = 5 μm. **b** Quantification of effects of SMIFH2 treatment on gametocyte morphology. Irregular parasites include those that are deformed or rounded in shape. The data represent the proportion of cells exhibiting each morphology. The means and standard error of the mean are shown from three independent experiments. **c** Immunofluorescence microscopy of SMIFH2 treated stage IV gametocytes labelled with anti-actin (red), anti-β-tubulin (magenta), native FRM2-GFP (green) and DAPI (blue). Concentrations of SMIFH2 are listed on the left. White arrows point to an accumulation of microtubules at the parasite periphery following SMIFH2 treatment. Blue arrows highlight a decrease in F-actin staining at the gametocyte tips. Scale bars = 5 μm. Source data for this figure is provided in the Source Data files.

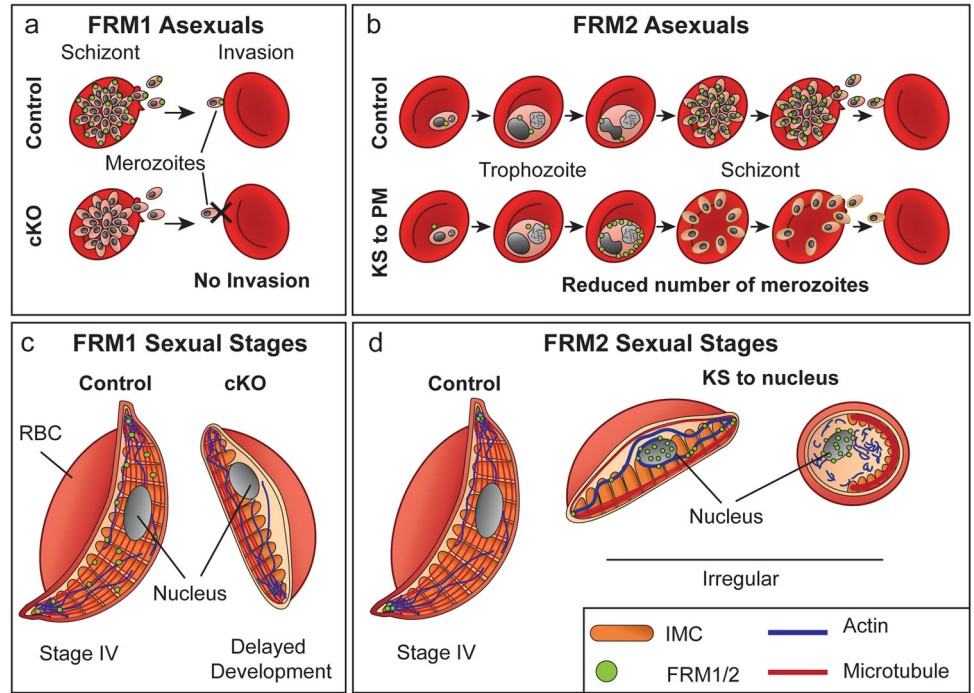

**Fig. 6 Schematic representation of the functions of formin proteins across blood-stage development. a** Asexual stage parasites express FRM1 late in asexual development where it locates at the apical end of the merozoites (green circles). Following conditional knockout (cKO) of FRM1, merozoites are unable to attach and invade RBCs. **b** FRM2 locates adjacent to the parasite nucleus (green circles), from early trophozoite stage through to merozoites within the schizont. Knock sideways (KS) of the FRM2 protein to the parasite plasma membrane (PM), leads to a reduction in merozoite numbers. **c** In gametocytes, FRM1 is distributed throughout the cytoplasm (green circles). Conditional knockout (cKO) of FRM1 in gametocytes leads to delayed development. **d** FRM2 locates at the tips of stage IV gametocytes (green circles) intertwined with the microtubule (red) and actin cytoskeletons (blue). Knock sideways (KS) of the FRM2 to the nucleus leads to a collapse of both the actin and microtubule networks and cells with irregular and round morphologies.

cell while in partially rounded cells the ends of the microtubule bundles appear to splay outwards. This is reminiscent of stage II-III gametocytes where the microtubule network is being laid down onto the developing IMC plates. Interestingly during the early stages of gametocyte development, FRM2 locates at the periphery of the cell at the IMC. Previous work has suggested that the IMC may be built through the trafficking of the membrane to the leading edges of the developing plates[39]. It is plausible that actin, and thus formins, play a role in this process.

As discussed in detail above, the accumulation of FRM2 at the tips of stage IV gametocytes may help stabilise the actin and tubulin networks. Upon transition to stage V of development, we observed loss of FRM2 from the tips and membrane of the gametocytes and an accumulation at a site near the food vacuole. It is interesting to speculate whether the removal of FRM2 from the tips of the stage IV gametocyte is the cause or the consequence of cytoskeleton disassembly (Fig. 6c, d). Further work is needed to test this question.

In this study, we show that both formin proteins are important for correct gametocyte development (Fig. 6c, d). Our data suggest that FRM1 is predominantly located in the gametocyte cytoplasm, where it may play a role in vesicle trafficking processes and organelle positioning (Fig. 6c). Previous work has shown that vesicular trafficking is important for the correct development and expansion of the IMC network and that disruption of this process leads to a delay in parasite development[39]. Similarly, treatment of gametocytes with the actin depolymerising agent, cytochalasin D, led to effects on parasite shape and dysregulation in mitochondria and apicoplast location[16]. Additional experiments are required to reveal the precise molecular function of FRM1.

Lastly, using the small molecular inhibitor, SMIFH2, we show that formin-based functions are targetable. SMIFH2 has been used extensively to study formins in other systems[34,35,43]. Here, we show that SMIFH2 treatment has a marked effect on asexual parasite development. The phenotypes of disrupted division and reinvasion recapitulate the phenotypes we observed upon FRM1 and FRM2 gene disruption. Likewise, SMIFH2 treatment of gametocytes prevents formation of the crescent falciform shape, recapitulating the phenotype observed when FRM2 is mis-localised. SMIFH2 lacks the specificity of action that would make it amendable to development as a drug; and additional work would be required to identify and develop formin inhibitors that could meet the criteria recommended by Medicines for Malaria Venture[44]. Nonetheless, our work points to the possibility of developing *Plasmodium*-specific reagents that target formin and actin-related processes. Such compounds are urgently needed to combat this important and deadly disease.

## Methods

**Parasite culture and transfection**. Asexual stage parasites were cultured in 0⁺ RBCs (Australian Red Cross blood service) at 5% haematocrit. Parasite-infected RBCs (iRBC) were cultured in RPMI-GlutaMAX-HEPES (Invitrogen) media supplemented with 5% (v/v) human serum (Australian Red Cross blood service), 0.25% AlbuMAX II (Invitrogen), 200 μM hypoxanthine (Sigma), 10 mM D-glucose (Sigma) and 20 μg/mL gentamycin (Invitrogen). Synchronous ring-stage cultures were obtained by incubating mixed-stage infected RBCs with 5% (w/v) D-sorbitol[45]. Late-stage asexual parasites and gametocytes were enriched from cultures, using magnetic separation[46]. Transfections were performed using standard electroporation methods and were maintained in culture media supplemented with either 5 nM WR99210, 1 μM DSM1 or 11 μM blasticidin S (BSD) depending on the selectable marker in

the plasmid[47]. High gametocyte-producing NF54 and 3D7 parasite lines were used in this work[48,49]. Two NF54 diCRE lines were used to generate FRM1-HA cell lines (see below); the first has the split CRE recombinase cassette integrated in the non-essential Rh3 locus and the second into the Pfs47 locus[30,50]. The NF54 diCRE (Rh3) was used for all experiments except for the Actin Chromobody reporter line which used the NF54-DiCre (Pfs47) line due to vector incompatibility issues. Asexual drug assays were performed in 96-well flat bottom plates and incubated for 48 h in the presence of drugs. Parasitemia's were counted using flow cytometry. Parasite-infected RBCs were stained with Syto61 (Invitrogen) at a final concentration of 200 nM in PBS for 45 min. Samples were analysed on a BD FACSCanto II Flow Cytometer System with an integrated BD High Throughput Sampler (HTS) using the APC filter sets. The flow cytometry data was analysed on the FlowJo v10 software (BD Biosciences)[36].

**Plasmid construction and PCR validation**. To create the pGLMS-FRM1-3xHA plasmid (referred to as FRM1-HA in this manuscript) a 5' homology sequence (HR1), artificial intron (containing a loxP site) and recondonised 3' of the gene was synthesised by Genewiz and cloned into the BglII and PstI sites of the pGLMS-HA plasmid. The 5' homology region corresponds to 7247–7647 bp of the FRM2 locus. The sequence from 7648–8025 bp was recodonised. The 3' homology region of FRM1 was PCR amplified using the FRM1-HR2-F and -R primers which contain EcoRI and KasI sites, respectively and directionally cloned into the pre-cut pGLMS-HA plasmid containing the synthesised sequence described above (Supplementary Table 1). The final plasmid was digested with BglI and BglII enzymes to destroy the vector backbone and generate the linear repair template for homology-driven repair. The CRISPR-Cas9 guide sequence (5'-GCTTCACACAATGGATCTGT-3', PAM site not shown) was selected using CHOPCHOP[51]. The guide was synthesised by Integrated DNA Technologies as an Alt-R CRISPR-Cas9 crRNA guide; duplexed with the Alt-R CRISPR-Cas9 tracerRNA (IDT, 1072532) and complexed with the Alt-R S.p. Cas9 Nuclease V3 (IDT, 1081058) following the manufactures instructions. This RNP complex was mixed with the linearised pGLMS-FRM1-HA plasmid and transfected as described above into NF54 parasites constitutively expressing a Rapamycin inducible split CRE recombinase (diCRE)[30].

To create the pSLI-FRM2-2xFKBP-GFP-2xFKBP plasmid (referred to as FRM2-GFP in this manuscript) a 705 bp fragment from the 3'end of the gene (minus the stop codon) was synthesised by GenScript and directionally cloned into the NotI and AvrII restriction sites of the plasmid. The final plasmid was transfected into 3D7 parasites as described above and the selection-linked integration performed[31]. Following transfection and drug selection, integration of the plasmid into the FRM2 locus was confirmed by PCR, using the FRM2-Int-F and GFP85-R primers (Supplementary Table 1 and Supplementary Fig. 1).

To create the Actin Chromobody reporter parasites, the propriety sequence for the Actin Chromobody (CB) was obtained from Chromotek under licence and a synthetic sequence was synthesised by Integrated DNA Technologies and directionally cloned into the NheI and SmaI sites of the pKIWI-BIP-GFP plasmid[52] to yield pKIWI-BIP-ActinCB-GFP. The pKIWI plasmid had been modified, replacing the 5' and 3' homology regions of P230p with the homology regions of the non essential gene PF3D7_0702500[53]. PF3D7_0702500 5' and 3' homology flanks were amplified from NF54 genomic DNA using the flank-specific primers (Supplementary Table 1). The CRISPR-Cas9 guide was chosen using CHOPCHOP[51] (5'-AAGATGACTTAA-GATCCATT-3', PAM site not shown) and cloned into pUF-Cas9

using InFusion. For transfection, 100 μg of pKIWI-BIP-ActinCB-GFP plasmid was linearise using SacI and XhoI and mixed with 50 μg of circular guide plasmid containing the PF3D7_0702500 guide sequence and transfected into synchronised NF54-DiCre (Pfs47) Formin-1-HA schizonts suspended in 100 μl of Nucleofection mix. Programme U33 with the Amaxa Basic Parasite Nucleofector Kit 2 (Lonza) was used. Integrated parasites were selected with 4 μl of blasticidin (10 mg/ml) for 5-days post transfection.

**Gametocyte cultures and stage progression assays**. Cultures containing synchronised gametocytes were generated using published methods[38,46]. Briefly, asexual stage parasites were cultured until they reached 8–10% trophozoite stage. At this point, they were expanded to obtain a culture at 2% parasitemia and 5% haematocrit. Spent culture media was left on these cultures at a ratio of 1:4 with fresh media to promote gametocyte commitment. (This represents day 0 in our assays). Culture medium was changed daily with the addition of N-acetyl-glucosamine to a final concentration of 62.5 mM (Sigma) to inhibit asexual growth. Parasite growth and gametocyte development were monitored by Giemsa-stained blood smears. If required, gametocytes were purified by magnetic separation[38,46].

For stage progression counts, parasite cultures were set up in triplicate and Giemsa-stained smears were made for each condition each day of the assay. Between fifty and one hundred gametocytes were observed and the stage determined based on previous studies[39]. These counts were then averaged across the three technical repeats and the proportion of each stage expressed as a percentage of the total gametocyte population. The data shown are the means from at least three independent experiments. The standard error of the mean is shown.

**diCRE-mediated conditional knockout of FRM1**. Infected ring-stage parasites were treated with either 100 nM rapamycin or DMSO (as the control) for asexual stage experiments and 250 nM Rapalog (A/C Heterodimeriser, Clontech) or ethanol (as the control) for gametocyte experiments. Parasites were treated with drug or mock treatment for 48 h to promote conditional gene knockout and samples were taken for PCR at this point. Parasites were returned to culture and grown to ~60 h post invasion to ensure all potential parasite invasion events had occurred and the parasitemia measured using flow cytometry following staining of the parasites with Syto61 as described above[36]. Assays were performed on 4 separate occasions, in triplicate for each repeat. The means for each experiment and the standard error of the mean is shown. Significance was determined by preforming a Student's t test. For gametocyte assays excision was performed as described above, with Rapalog added at day 0 when gametocyte rings are present and replaced daily for 2 days. Gametocytes were cultured for the remainder of the assay, and counts were performed as described above.

**Knock-sideways studies and growth assays**. For the knock-sideways experiments, 3D7-FRM2-GFP parasites were transfected with either the mislocalizer plasmid pLyn-FRB-mCherry (BSD) for asexual stage experiments or pNLS-FRB-mCherry (DSM1) for gametocyte experiments[31]. The pLyn-FRB-mCherry construct allows mislocalisation towards the parasite plasma membrane (PPM) while the NLS construct allows for mislocalisation to the nucleus. Mislocalisation was verified by live-cell microscopy. To facilitate ongoing FRM2-GFP mislocalisation the parasites were cultured continuously in the presence of 250 nM Rapalog for the duration of the assays.

Asexual growth assays were performed over multiple parasite cycles, using the plasma membrane mislocaliser parasites. Assays were initiated at 1% ring stage (cycle 1) and allowed to reinvade. Following reinvasion (cycle 2) the parasitemia was measured using flow cytometry as described above and the parasitemia for control and treated groups recorded. The cultures were then equally diluted based on the dilution factor required to obtain a 1% culture in the control groups. The culture was grown for another cycle, and the parasitemia was measured (cycle 3). The data plotted is the cumulative parasitemia whereas the parasitemia of cycle 3 equals the sum of the cycle 1, 2 and 3 parasitemia. Each experiment was performed in triplicate on 3 separate occasions. The mean and the standard error of the mean are shown. The significance at each time point is calculated using a Student's $t$ test. Gametocyte assays were performed with the nuclear mislocaliser parasites, with gametocytes cultured in the presence of 250 nM Rapalog or ethanol for the duration of the assays. Counts were performed as described above.

**SMIFH2 treatment**. For asexual stage experiments, either dihydroartemisinin or the Small Molecule Inhibitor of Formin Homology domain 2 (SMIFH2) at a range of concentrations were incubated with ring-stage asexual stage parasites for 1 cycle, prior to assessment of the parasitemia by flow cytometry as described above[36].

For SMIFH2 gametocyte experiments, Assays were set up as described for the knock-sideways experiments. The Small Molecule Inhibitor of Formin Homology domain 2 (SMIFH2) was prepared fresh in DMSO for each experiment. Developing gametocytes from day 0 were incubated with different concentrations of SMIFH2 ranging from 12.5 to 100 µM. DMSO was used as a control. Counts were performed as described above.

**Western blots**. Synchronised late asexual schizont cultures were enriched using a CS magnetic column (Miltenyi Biotech). Enriched schizont-infected erythrocytes were saponin lysed, washed with phosphate-buffered saline (PBS), and resuspended in 4× Laemmli's buffer. To detect FRM2, proteins were separated on NuPAGE 3–8% Tris-acetate polyacrylamide gels with Tris-acetate buffer (Invitrogen), transferred to nitrocellulose membrane with the addition of 0.1% SDS, blocked in 5% (w/v) PBS- 0.1% Tween 20 and probed with primary antibody rabbit anti-FKBP (Abcam, Ab24373, 1:2000) and followed by rabbit HRP-conjugated secondary antibody (Merck, 1:4000) and the signal was detected using Clarity Western ECL Substrate (BioRad). To detect Aldolase, proteins were separated on NuPAGE 4–12% Bis-Tris polyacrylamide gels with MOPS Buffer (Invitrogen), transferred to nitrocellulose membrane, blocked in 5% (w/v) PBS- 0.1% Tween 20 and probed with primary antibody rabbit anti-Aldolase (1:1000[54]) and followed by rabbit HRP-conjugated secondary antibody (Merck, 1:4000) and detected as described above.

**Live-cell and immunofluorescence microscopy**. Live-cell microscopy of the FRM2-GFP cell line was performed using a DeltaVision Elite Restorative Widefield Deconvolution System (GE Healthcare) and a 100x UPLS Apo (1.4NA) objective lens under oil immersion (GE Healthcare). Samples were taken from the culture and mounted under a coverslip. Gametocytes were stained with BODIPY-TR-Ceremide lipid dye (1 µM final) (Thermo Fisher) at 37 °C overnight.

Immunolabelling was performed in a dark humidified chamber. Coverslips were coated with 0.1 mg/mL PHAE (erythroagglutinating phytohemagglutinin, Sigma) in PBS for 20 min at 37 °C. Infected RBCs were harvested by centrifugation, washed in 1× PBS and settled on the PHAE-coated coverslips at 37 °C for 10 min. Unbound cells were washed off with 1× PBS. Cells were fixed in either 2% formaldehyde/0.0065% glutaraldehyde or 2% formaldehyde/0.0025% glutaraldehyde, in microtubule stabilising buffer (100 mM 2(N-morpholino)ethanesulfonic acid, 1.5 M NaCl2, 50 mM ethylene glycol-bis(beta-aminoethylether)tetraacetic acid), 50 mM glucose, 50 mM MgCl2[55] for 10 to 20 min at room temperature. Wells were washed three times with 1× PBS. Permeabilization was carried out for 20 min using 0.2% Triton-X100/PBS. Wells were washed with 1× PBS and blocked in 3% BSA/PBS for 1 h. Primary antibodies (in 3% BSA/PBS) were applied and incubated for 1 h at room temperature. The following primary antibodies were used: rabbit anti-actin1.1 (1:300[22]), mouse anti-β-tubulin (clone TUB 2.1, 1:300, Sigma Aldrich), mouse anti-GFP (clones 7.1 and 13.1, 1:300, Roche), mouse anti-centrin (1:100, clone 20H5 mouse, Millipore), rabbit anti-GAP45 (1:500[54]), rabbit anti-AMA1 (1:250, EF24, Robin Anders), rabbit anti-EBA175 (1:500, Ab 1552[56]), rabbit anti-ACP (1:500[57]), rabbit anti-GFP (1:300[58], rabbit anti-PhIL1 (1:300[39], chicken anti-GFP (1:500, Invitrogen), mouse anti-HA (clone HA-7, 1:500, Sigma Aldrich), mouse anti-HA-Biotin (clone HA-7, 1:500, Sigma Aldrich). After washing, secondary antibodies (1:400 in 3% BSA/PBS) were applied for 1 h. Unbound antibodies were washed off. Secondary antibodies: anti-mouse and anti-rabbit IgG-conjugated Alexa Fluor 488, 568 and 647; anti-chicken IgY Alexa Fluor 647; Streptavidin-Alexa Fluor 488 (Life Technologies). In total, 2 µg/mL DAPI working solution was applied and incubated for 10 min. After washing, samples were mounted in 90% glycerol-containing anti-fading agent p-phenylenediamine (0.2% w/v) in PBS, pH 8.6. Slides were sealed and stored at 4 °C. In some immunolabelling experiments, actin filaments were stabilised by incubation of iRBCs with 1 µM Jasplakinolide for 30 min to 2 h at 37 °C prior to preceding with immunolabeling[16].

Microscopy was performed using a DeltaVision Elite Restorative Widefield Deconvolution System (GE Healthcare) and a 100x UPLS Apo (1.4NA) objective lens under oil immersion (Applied Precision). Samples were excited with solid-state illumination (Insight SSI, Lumencor). The following filter sets with excitation and emission wavelengths were used: DAPI Ex390/18, Em435/48; FITC, Ex475/28, Em523/26; TRITC, Ex542/27, Em594/45; Cy5 Ex 632/22, 676/34 nm. Z-stacks (0.2-µm steps) were deconvolved using the default settings in the SoftWoRx 5.0 (GE Healthcare) acquisition software. For 3D-SIM imaging, the DeltaVision OMX V4 Blaze was used (GE Healthcare). Samples were excited using 488- or 568-nm lasers and imaged using bandpass filters at 528/48 and 609/37 with a 60X Olympus Plan APO N (1.42 NA) oil immersion lens. Images were further processed using FIJI ImageJ software (version 2.0.0-rc-65/1.52c).

**Live-cell microscopy invasion assays**. Synchronous ring-stage FRM1-HA parasites received either DMSO as the mock control (0.05%) or Rapamycin (Rap, 100 nM) and were incubated for approximately 48 h until they reached schizont stage. Here, they were mounted in an environmental chamber where they were filmed at 4 s per frame as using the AxioCam 702 Mono camera on a Zeiss Cell observer widefield microscope[32,33]. Video files were manipulated and labelled in FIJI and invasion data was graphed in GraphPad Prism.

**Statistics and reproducibility**. All experiments in this work were repeated at least three times and the data is represented as means plus or minus the standard deviation. Detailed methods are provided to unsure the reproducibility of experiments. Cell numbers and replicates are noted in each figure legend as it the

statistical test used and the significance. Statistical analyses were conducted using GraphPad Prism 8.

**Reporting summary**. Further information on research design is available in the Nature Portfolio Reporting Summary linked to this article.

## Data availability

Additional data are available in Supplementary Information. Source data for all graphs and plots in the paper can be found in supplementary data files 1, 2 and 3. The datasets generated and analysed during the current study are available from the corresponding authors on reasonable request.

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

## Acknowledgements

We acknowledge the traditional custodians of the lands on which this project was conducted, the Wurundjeri people of the Kulin Nation. We thank Prof Jake Baum, The University of New South Wales for PfGAP45 antibodies and for providing the pKIWI constructs. We thank Professor Alan Cowman, Walter and Eliza Hall Institute for providing anti-RON4 and anti-EBA175 antibodies and Robin Anders for providing anti-AMA1 antibodies. We acknowledge the Biological Optical Microscopy Platform, The University of Melbourne, for microscopy assistance. L.T. is a Georgina Sweet, Australian Research Council Laureate Fellow (FL150100106) (http://www.arc.gov.au). M.W.A.D. and L.T. thank the National Health and Medical Research Council (1098992) (https://www.nhmrc.gov.au) for funding this work. The authors thank the Australian Red Cross Blood Service.

## Author contributions

Conceptualisation: S.C., L.T. and M.W.A.D.: investigation: S.C., E.P., M.D., D.L., T.A.T., S.L., D.S.M., M.A.S., D.A., S.T., P.J.M. and M.W.A.D.: formal analysis: S.C., M.D., D.L., P.G. and M.W.A.D.: writing—original draft: S.C., L.T. and M.W.A.D.: writing—reviewing and editing: S.C., E.P., M.D., D.L., T.A.T., S.L., D.A., S.T., P.J.M., P.G., L.T. and M.W.A.D.: visualisation: S.C., E.P., M.D., D.L., T.A.T., P.J.M., P.G. and M.W.A.D.: supervision: P.G., L.T. and M.W.A.D.: funding acquisition: L.T. and M.W.A.D.

## Competing interests

The authors declare no competing interests.
