## [Peer Review File · Communications Biology]

Reviewers' comments:

Reviewer #1 (Remarks to the Author):

This is a well conducted and beautifully illustrated manuscript investigating the roles of two actin filament forming proteins in *Plasmodium falciparum* blood stages. The clearly presented data reveal important new insights, add to previous published work and also contribute to some controversy. I have only minor suggestions to improve an already nearly perfect manuscript.

Line 30/31: please use only 2 meaningful digits, e.g. 240 million instead of 241 and 630.000 instead of 627.000, even though WHO presents them as stated.

In intro, please distinguish between *Plasmodium* and *P. falciparum*. I would suggest to only use *P. falciparum*, if the available data only refers to this parasite, but use *Plasmodium*, when work in e.g. rodent model parasites also showed importance of the proteins, which the authors partly do. It could make it a more complete read, if also work on other stages is briefly mentioned, e.g. actin II in oocysts and actin I in sporozoites. Work in *T. gondii* might be referred to as well in slightly more depth, especially as formins were extensively investigated.

Reviewer #2 (Remarks to the Author):

Plasmodium, as all Apicomplexa, has a limited set of actin-binding proteins and formins are the only F-actin nucleator found in these parasites. Considering the essential role of F-actin for many cellular processes, especially motility and invasion in Apicomplexa, understanding the role of formins is of special interest.

In this study, Collier and colleagues investigate the role of the two formins encoded in *Plasmodium falciparum* genome, FRM1 and FRM2, through conditional knock-down systems.

They confirm published data about the localization of FRM1 and FRM2 in the asexual stages and also the phenotype of FRM2 in these stages although it is less pronounced likely due to the leakiness of the strategy used to knock-down FRM2 (knock sideways).

Additionally, they generated a FRM1-cKO cell line and show that the merozoites are likely impaired in invasion and they describe the localization of FRM1 and FRM2 in gametocytes and assess their role along gametocyte development.

Overall, the work is well-conducted and the experiments and results are well described with the appropriate controls. However, some major concerns, listed below, need further investigation/explanation before publication.

Major concerns

1. The results are a bit difficult to follow. I understand that the authors choose to describe first the localization of the formins and then the phenotypes but we navigate from the RBC stages of FRM1 and then FRM2 to the gametocytes for the localization to come back to the RBC stages and finally the gametocytes of FRM1 and FRM2 for the phenotypes. I suggest to rearrange the paragraphs so that they describe first the RBC stages (localization and functions) and then gametocytes (localization and functions) as it is done in the discussion as well as in the final model (Figure 6).

2. lines 244-245, the authors suggest that FRM1 may be required for the formation or release of the rhoptries and micronemes. However, they mentioned earlier that the staining of RON4 was not change upon FRM1 excision, suggesting that the rhoptries (at least) are made and well docked at the apical pole. Could the author also check for the micronemes by immunofluorescence? Would it be possible to

evaluate if the parasites attach to the RBC but fail to enter by giemsa staining or by immunofluorescence?

Since FRM1 is involved in actin polymerization, it is most likely a motility defect that prevent the parasites to be dispersed and invade new RBCs. Would it be possible to stain F-actin in the FRM1-HA strain with the actin chromobodies? The apical dot of F-actin is probably not there anymore.

3. Concerning the knock sideways of FRM2 in the RBC stages, the authors noted "a less dramatic effect on growth than the previously reported KO defect" (lines 259-260) but do not show the fate of the apicoplast in their strain. The fragmentation of the apicoplast was the main defect noticed upon FRM2-KO.

The co-localization of FRM2 with centrin is interesting but needs a little bit more comment. Lines 265-267: there is no explanation for the quantification of the co-occurrence of centrin and FRM2. Does it mean that the knock sideways is just partially efficient because a part of FRM2 still co-localizes with centrin, knowing that the centrosome positioning is important for the apicoplast segregation? Do the authors assessed the number of centrin dots vs number of nuclei?

4. In the text, lines 294-295, it is mentioned that the mCherry signal of the NLS mislocaliser goes to the nucleus but in Fig 4a, Fig. S6a-c, and Fig S7, (i) the signal seems larger than the DAPI and (ii) there is also an additional (strong) signal that seems to be peripheral. Is it the case? To my view, it should also be mentioned as it contributes to prevents the polarized signal of FRM2 at the poles. Therefore, the deformation (tear-drop and rounded parasites) is likely due to the mislocalized FRM2 that polymerized actin at the periphery of the parasite rather than at the poles. It would fit with the IFA presented in Fig. 4d-e and Fig 7S where we see mislocalized actin and microtubule cytoskeletons rather that disassembly. Did the authors considered this interpretation?

Minor point

1. FRM1 and FRM2 are often presented in the same figures so, for clarity, it would be nice if the FRM name is always written in the panels as it is well done in Figures 1 and 2. For example, FRM1-HA should be written in panel a of figure 3 and FRM1-HA and FRM2-GFP in panels a and e, respectively. Maybe it would also be possible to add this information on the graphs.
2. The panel a of Supplementary Fig. 2. could be a main figure since it is the first description of FRM1 localization in gametocytes.
3. Panels a to c of Supplementary Fig. 5. could also be a main figure since it is the first description of FRM1 phenotype in gametocytes.
3. The time in video S1 is not readable (cut).

Reviewer #3 (Remarks to the Author):

The authors investigate two formins in *Plasmodium falciparum* for their function and potential as drug targets. Using complex imaging and conditional knockout or knock sideways lines, they show that FRM1 and FRM2 are essential for parasite survival although they have different roles (merozoite invasion vs cell division). They also show that targeting formin proteins with a formin inhibitor results in multistage parasite growth defects. The results are novel and will be of interest to others in the community.

It would have been interesting for the authors to discuss other aspects that make a protein a "good" drug target, such as resistance generation and structure/assay availability.

The authors could consider updating malaria statistics to the 2022 World Malaria Report.

The methods are very well described and the supplementary information is comprehensive. The figures and discussions are of a high quality.

Response to reviewers

We thank the reviewers for their positive comments about our manuscript and thank them for their suggestions for improving our work. Please find below a detailed response to all the questions and comments from the reviewers.

Reviewer #1 (Remarks to the Author):

Line 30/31: please use only 2 meaningful digits, e.g. 240 million instead of 241 and 630.000 instead of 627.000, even though WHO presents them as stated.

As requested by the reviewer we have edited the WHO numbers. We have also updated these to the most recent 2022 numbers as suggested by reviewer 3.

In intro, please distinguish between Plasmodium and P. falciparum. I would suggest to only use P. falciparum, if the available data only refers to this parasite, but use Plasmodium, when work in e.g. rodent model parasites also showed importance of the proteins, which the authors partly do.

We have modified all Plasmodium to be species specific where appropriate, this includes modifying the title.

It could make it a more complete read, if also work on other stages is briefly mentioned, e.g. actin II in oocysts and actin I in sporozoites.

We have modified the text to include the work from oocysts and sporozoites.

Text has been added describing the role of actin I in ookinetes and sporozoites (Lines 55 - 56)

Additional text has been added highlight the importance of actin II in ookinete conversion, oocyst development and sporogony (Lines 67 – 71)

Work in T. gondii might be referred to as well in slightly more depth, especially as formins were extensively investigated.

We have added text to the introduction highlighting the work previously performed on the three formin proteins found in *Toxoplasma gondii*. (Lines 99-103)

Reviewer #2 (Remarks to the Author):

Major concerns

1. The results are a bit difficult to follow. I understand that the authors choose to describe first the localization of the formins and then the phenotypes but we navigate from the RBC stages of FRM1 and then FRM2 to the gametocytes for the localization to come back to the RBC stages and finally the gametocytes of FRM1 and FRM2 for the phenotypes. I suggest to rearrange the paragraphs so that they describe first the RBC stages (localization and functions) and then gametocytes (localization and functions) as it is done in the discussion as well as in the final model (Figure 6).

We appreciate the reviewers' comments. To make the results section easier to follow, we have rearranged the figures and paragraphs so that the localisation and phenotypes of the different lifecycle stages are described together, as per the reviewer's suggestion.

2. lines 244-245, the authors suggest that FRM1 may be required for the formation or release of the rhoptries and micronemes. However, they mentioned earlier that the staining of RON4 was not change upon FRM1 excision, suggesting that the rhoptries (at least) are made and well docked at the apical pole. Could the author also check for the micronemes by immunofluorescence?

We have now conducted IFAs using antibodies against the microneme proteins AMA1 and EBA175 (Supplementary Figure 2B & C). From these IFAs, it does not look as though there is any aberrant phenotype associated for either protein upon excision of FRM1. Whilst the protein appears to be more densely-packed in some regions of the cKO parasites compared to the control, we still see both AMA1 and EBA175 localising to the apical tips of the merozoite as expected. This suggests that like the RON4 labelled rhoptries, the micronemes are still able to form correctly in the absence of FRM1.

We have added reference to these IFAs in the text (Line 156-157). In addition, we have modified the discussion surrounding this, to remove the speculation about microneme localisation following FRM1 deletion. (Line 419-421).

Would it be possible to evaluate if the parasites attach to the RBC but fail to enter by giemsa staining or by immunofluorescence?

In our work we have used live cell imaging to visualise and quantitate invasion events, which is the gold standard experiments for investigating this process. As explained in the text, we see merozoites contacting RBCs and some evidence for weak deformation of the RBCs. We observed no attachment of the merozoites to the RBC, as is observed in invasion events in control samples.

Since FRM1 is involved in actin polymerization, it is most likely a motility defect that prevent the parasites to be dispersed and invade new RBCs. Would it be possible to stain F-actin in the FRM1-HA strain with the actin chromobodies? The apical dot of F-actin is probably not there anymore.

We have now made transgenic parasites expressing a GFP actin Chromobody reporter in the FRM1-HA line. The actin chromobody line shows a similar labelling pattern to FRM1-HA. F-actin is evident at the apical ends of the daughter merozoites developing within late-stage schizonts. We also observed polymerised actin accumulating adjacent to the food vacuole in many cells. These results are similar to those described by Stortz *et al.* (2019). Upon cKO of FRM1, F-actin appears to be largely lost from the merozoite tips and becomes diffusely distributed throughout the cytoplasm, associated with brighter stabilised filament-like structures. We have added images from this work to Supplementary figure 2d and have added additional text to this section (Lines 160– 167). A full description of the generation of this cell line has been added to the methods.

3. Concerning the knock sideways of FRM2 in the RBC stages, the authors noted “a less dramatic effect on growth than the previously reported KO defect” (lines 259-260) but do not show the fate of the apicoplast in their strain. The fragmentation of the apicoplast was the main defect noticed upon FRM2-KO.

We have now conducted IFAs, using an anti-ACP antibody, to investigate the fate of the apicoplast following mislocalisation of FRM2. We observed an intermediate phenotype in aberrant schizonts compared to the severely disrupted examples provided by Stortz *et al.* (2019). In most cases, an apicoplast is distributed to the daughter merozoite; however, larger apicoplast puncta are observed in some cells. Thus, it appears that the apicoplast is still segregated correctly, but the number is reduced, in line with the reduction in the number of nuclei. This data has been added to Supplementary fig 2j and text has been added to the Results (Lines 239-242). The differences in our results and the published work are addressed in the Discussion.

The co-localization of FRM2 with centrin is interesting but needs a little bit more comment. Lines 265-267: there is no explanation for the quantification of the co-occurrence of centrin and FRM2. Does it mean that the knock sideways is just partially efficient because a part of FRM2 still co-localizes with centrin, knowing that the centrosome positioning is important for the apicoplast segregation? Do the authors assessed the number of centrin dots vs number of nuclei?

We apologise if the FRM2/centrin experiments were not clearly described.

We have now added a statement about the efficiency of the FRM2-GFP knock sideways (Line 224-227). We think that FRM2 that is already located at the centrosome may be resistant to Rapamycin-induced redistribution to the nucleus, most probably due to the high affinity of the FRM2-actin complex. The knock-sideways system is likely more efficient in capturing proteins as they are produced. We have modified the text to better explain the rationale for the centrin experiments and the use of the Pearson's co-efficient for measuring co-occurrence (Lines 230-238). We have added to text to the Discussion (Lines 430-434). We do not assess the number of centrin puncta relative to the number of nuclei. Due to the asynchronous nature of asexual division in Plasmodium, there may be occasions where a nuclei appears to have 2 centrin puncta due to that particular nuclei being in the process of dividing.

4. In the text, lines 294-295, it is mentioned that the mCherry signal of the NLS mislocaliser goes to the nucleus but in Fig 4a, Fig. S6a-c, and Fig S7, (i) the signal seems larger than the DAPI and (ii) there is also an additional (strong) signal that seems to be peripheral. Is it the case? To my view, it should also be mentioned as it contributes to prevents the polarized signal of FRM2 at the poles. Therefore, the deformation (tear-drop and rounded parasites) is likely due to the mislocalized FRM2 that polymerized actin at the periphery of the parasite rather than at the poles. It would fit with the IFA presented in Fig. 4d-e and Fig 7S where we see mislocalized actin and microtubule cytoskeletons rather than disassembly. Did the authors considered this interpretation?

We thank the reviewer for these comments.

The NLS reporter is a soluble protein and will occupy the entire nuclear compartment, whereas DAPI only labels the DNA itself. The accumulation of the FRM2-GFP signal at the nuclear periphery, following induction of the mislocalisation event, is interesting. The large size of the complex of the FRM2-2xFKBP-GFP-2XFKBP fusion and the FRB-mCherry mislocaliser, may exceed the maximum size for ready passage through the nuclear pores. Thus, much of the protein may remain attached to NLS binding sites on the cytoplasmic site of the nuclear membrane.

With respect to the peripheral staining that is sometimes observed, we hypothesise that some of the soluble FRB-NLS-mCherry may bind to membrane/cytoskeleton bound FRM2-GFP at the periphery of the parasite. In this case the presence of an NLS tag may not be sufficient to displace the already bound and localised FRM2-GFP chimera to the nucleus.

We agree with your suggestion that some of the localisation patterns and shapes may be a result of the mislocalised FRM2-GFP binding and nucleating actin in the wrong place.

We have added text to the results section (Lines 329-332) and to the discussion (Line 426-430) to address this.

Minor point

1. FRM1 and FRM2 are often presented in the same figures so, for clarity, it would be nice if the FRM name is always written in the panels as it is well done in Figures 1 and 2. For example, FRM1-HA should be written in panel a of figure 3 and FRM1-HA and FRM2-GFP in panels a and e, respectively. Maybe it would also be possible to add this information on the graphs.

We have added these FRM1 and FRM2 labels to the panels as required.

2. The panel a of Supplementary Fig. 2. could be a main figure since it is the first description of FRM1 localization in gametocytes.

We have considered moving these supplementary figures to the main figures, however due to figure number constraints and space we have left these in the supplementary

3. Panels a to c of Supplementary Fig. 5. could also be a main figure since it is the first description of FRM1 phenotype in gametocytes.

As above, we would have liked to include these figures in the main figures, but are unable to accommodate them due to space constraints.

3. The time in video S1 is not readable (cut).

We have fixed the time stamp on video S1.

Reviewer #3 (Remarks to the Author):

It would have been interesting for the authors to discuss other aspects that make a protein a "good" drug target, such as resistance generation and structure/assay availability.

We thank the Reviewer for this suggestion. We feel that an extensive discussion of the characteristics of good drug targets is beyond the scope of the Discussion for this manuscript.

We have modified the discussion to: "SMIFH2 lacks the specificity of action that would make it amendable to development as a drug; and additional work would be required to identify and develop formin inhibitors that

could meet the criteria recommended by Medicines for Malaria Venture (Burrows et al. 2017). Nonetheless, our work points to the possibility of developing *Plasmodium* specific reagents that target formin and actin-related processes. Such compounds are urgently needed to combat this important and deadly disease.”

The authors could consider updating malaria statistics to the 2022 World Malaria Report.

As suggested by Reviewer #1, we have updated the malaria statistics.

REVIEWERS' COMMENTS:

Reviewer #2 (Remarks to the Author):

The authors were very responsive and addressed all my concerns.
The article is clear and of high quality. It will be of great interest for the community.

Response to reviewers

We gain thank all the reviewers for their positive comments about our manuscript and thank them for their suggestions for improving our work. Please find below a detailed response to all the questions and comments from the reviewers.

REVIEWERS' COMMENTS:

Reviewer #2 (Remarks to the Author):

The authors were very responsive and addressed all my concerns.
The article is clear and of high quality. It will be of great interest for the community.

Reviewer 2 had no further concerns to address.